# Machine Learning Detects Intraventricular Haemorrhage in Extremely Preterm Infants [note 1]

**DOI:** 10.3390/children10060917

**Published:** 2023-05-23

**Authors:** Minoo Ashoori, John M. O’Toole, Ken D. O’Halloran, Gunnar Naulaers, Liesbeth Thewissen, Jan Miletin, Po-Yin Cheung, Afif EL-Khuffash, David Van Laere, Zbyněk Straňák, Eugene M. Dempsey, Fiona B. McDonald

**Affiliations:** 1INFANT Research Centre, University College Cork, T12 AK54 Cork, Ireland; 120224294@umail.ucc.ie (M.A.); jotoole@ucc.ie (J.M.O.); k.ohalloran@ucc.ie (K.D.O.); g.dempsey@ucc.ie (E.M.D.); 2Department of Physiology, School of Medicine, College of Medicine and Health, University College Cork, T12 XF62 Cork, Ireland; 3Department of Paediatrics and Child Health, School of Medicine, College of Medicine and Health, University College Cork, T12 DC4A Cork, Ireland; 4Department of Development and Regeneration, Katholieke Universiteit Leuven, Herestraat 49, 3000 Leuven, Belgium; gunnar.naulaers@uzleuven.be; 5Neonatal Intensive Care, Katholieke Universiteit Hospital Leuven, Herestraat 49, 3000 Leuven, Belgium; liesbeth.thewissen@uzleuven.be; 6Paediatric and Newborn Medicine, Coombe Women’s Hospital, D08 XW7X Dublin, Ireland; jmiletin@coombe.ie; 7Department of Paediatrics, University of Alberta, Edmonton, AB T6G 1C9, Canada; poyin@ualberta.ca; 8Faculty of Medicine and Health Sciences, Royal College of Surgeons in Ireland, D02 P796 Dublin, Ireland; afifelkhuffash@rcsi.com; 9Neonatale Intensive Care Unit, Universitair Ziekenhuis, (UZ) Antwerp, Drie Eikenstraat 655, 2650 Antwerp, Belgium; david.vanlaere@uza.be; 10Institute for the Care of Mother and Child, Third Faculty of Medicine, Charles University, 100 00 Prague, Czech Republic; z.stranak@seznam.cz

**Keywords:** near-infrared spectroscopy (NIRS), regional cerebral oxygen saturation (rcSO_2_), peripheral oxygen saturation (SpO_2_), prolonged relative desaturation (PRD), extreme gradient boosting (XGBoost)

## Abstract

Objective: To test the potential utility of applying machine learning methods to regional cerebral (rcSO_2_) and peripheral oxygen saturation (SpO_2_) signals to detect brain injury in extremely preterm infants. Study design: A subset of infants enrolled in the Management of Hypotension in Preterm infants (HIP) trial were analysed (*n* = 46). All eligible infants were <28 weeks’ gestational age and had continuous rcSO_2_ measurements performed over the first 72 h and cranial ultrasounds performed during the first week after birth. SpO_2_ data were available for 32 infants. The rcSO_2_ and SpO_2_ signals were preprocessed, and prolonged relative desaturations (PRDs; data-driven desaturation in the 2-to-15-min range) were extracted. Numerous quantitative features were extracted from the biosignals before and after the exclusion of the PRDs within the signals. PRDs were also evaluated as a stand-alone feature. A machine learning model was used to detect brain injury (intraventricular haemorrhage-IVH grade II–IV) using a leave-one-out cross-validation approach. Results: The area under the receiver operating characteristic curve (AUC) for the PRD rcSO_2_ was 0.846 (95% CI: 0.720–0.948), outperforming the rcSO_2_ threshold approach (AUC 0.593 95% CI 0.399–0.775). Neither the clinical model nor any of the SpO_2_ models were significantly associated with brain injury. Conclusion: There was a significant association between the data-driven definition of PRDs in rcSO_2_ and brain injury. Automated analysis of PRDs of the cerebral NIRS signal in extremely preterm infants may aid in better prediction of IVH compared with a threshold-based approach. Further investigation of the definition of the extracted PRDs and an understanding of the physiology underlying these events are required.

## 1. Introduction

Survival in the preterm population is improving, especially for extremely preterm infants. However, long-term adverse neurodevelopmental outcomes remain an ongoing concern, with morbidity increasing with decreasing gestational age [1,2,3]. A recent systematic review suggests that between 5–52% of preterm infants will develop some degree of intraventricular haemorrhage (IVH); the disparity attests to the variability in clinical practice and potential diversity within the patient population [4]. Many studies have associated IVH in extremely preterm infants with poor neurodevelopment in the early years of life [5,6]. A recent study examined children who were born extremely preterm but were assessed during school years (age 8) and reported that all grades of IVH were associated with a higher risk of cerebral palsy and higher grades of IVH were associated with a higher risk of impaired academic scores [7].

The dose and severity of the physiological insults or intrinsic vulnerability corresponding to the different grades of IVH are unknown [8]. Matsushita et al. (2021) performed a retrospective study examining potential similarities using 62 clinical features from 215 individual infants born <1000 g [9]. The cluster analysis revealed six distinct phenotypic clusters, some of which were associated with IVH. The authors’ unsupervised machine learning approach may contribute to a better understanding of the IVH population in the future [9]. Hypoxia is believed to be a contributory factor in the development of IVH. A recent study found that infants with IVH spent a greater percentage of time with a heavier burden of cerebral hypoxia, compared with those who did not have IVH [10]. This is congruent with changes in systematic and cerebral haemodynamics that reveal a pattern consistent with a hypoperfusion–reperfusion cycle [11]. Impaired cerebral autoregulation may result in cerebral blood flow disturbance and potential rupture of the fragile vasculature of the germinal matrix, resulting in IVH [12,13,14]. 

Near-infrared spectroscopy (NIRS) is a noninvasive optical technology that is used for continuous bedside monitoring of regional cerebral oxygen saturation (rcSO_2_) and provides information on neonatal cerebral haemodynamics [15]. Regional cerebral oxygenation may be an important tool to guide treatment and prevent cerebral hypoxia [16] in the most immature infants [15]. Cerebral NIRS has become commonplace in clinical practice [17,18]; however, a recent meta-analysis, including data from 2606 infants concluded that there were insufficient data to support or reject the benefit of cerebral NIRS monitoring for improved clinical outcome measures [19]. This is not surprising as there is no general agreement concerning the absolute values of rcSO_2_ associated with brain damage. These absolute values have ranged from rcSO_2_ < 50% [20,21] to rcSO_2_ < 65% [22], and in one recent randomised trial there was no association between rcSO_2_ < 55% or rcSO_2_ > 85% and adverse long-term outcomes [17]. The variability in absolute values of rcSO_2_ between NIRS devices and sensors and between different gestational ages makes it difficult to reach a consensus on normal values of rcSO_2_ in preterm infants [22,23]. Therefore, alternative methods of describing the continuous signal should be evaluated. 

We have previously employed signal processing methods to investigate the association of cerebral oxygenation in preterm infants born less than 32 weeks of GA with IVH and periventricular leukomalacia (PVL). Features of the signal amplitude within certain frequency bands (0.9–3.6 mHz) were useful to detect brain injury [24]. Subsequently, we detected the presence of desaturation waveforms in rcSO_2_ in extremely preterm NIRS recordings and reported that removing these prolonged relative desaturations (PRDs; data-driven desaturations) increased the predictability of the NIRS signals in detecting brain injury [25]. However, this was based on a relatively small sample size (*n* = 10). The objective of the current study was to isolate PRDs in rcSO_2_ and SpO_2_ signals in preterm infants less than 28 weeks and using machine learning models, explore their putative relationship with brain injury in preterm infants. Furthermore, we sought to examine the potential value of combining clinical features with features from the NIRS signal to detect preterm brain injury.

## 2. Methods

### 2.1. Study Design and Participants

The current study analysed data derived from a multicentre clinical trial, the Hypotension in Preterm Infants (HIP, Trial registration number NCT01482559, EudraCT 2010-023988-17) [26]. The infants were recruited between February 2015 and September 2017 at Cork University Maternity Hospital. All infants were extremely preterm, i.e., born less than 28 weeks’ gestation. Data from 46 extremely preterm infants were analysed in this study, including both male (*n* = 25) and female (*n* = 21) infants. The mean birth weight of the infants was 767.41 g ± 155.57 g, 20 of whom were diagnosed with hypotension. All infants underwent continuous (>24 h) rcSO_2_ measurements using NIRS over the first 72 h and cranial ultrasounds performed during the first week of life. A subset of these infants (*n* = 32) also had continuous SpO_2_ data available. This study was approved by the Clinical Research Ethics Committee, Cork, ECM 5(2) 15 January 2013.

### 2.2. Data Collection

The INVOS 5100 device, (Covidien Mansfield, MI, USA) was used with the neonatal transducer, INVOS OxyAlert NIRSensor (Covidien, Mansfield, MI, USA) to make continuous measurements of rcSO_2_. The cerebral sensor was placed on the right frontotemporal area of the forehead, measuring the ratio of oxygenated haemoglobin to the total haemoglobin in the tissue beneath. When available, the Moberg CNS device (Moberg, PA, USA) was used to time-synchronise and store the rcSO_2_ and peripheral oxygen saturation (SpO_2_) data.

SpO_2_ was recorded using a pulse oximeter (IntelliVue, MP70, Philips Healthcare, Best, The Netherlands, or equivalent). SpO_2_ data were only available for the infants whose rcSO_2_ data was stored using a connected device (Moberg Neuroscan, Ambler, PA, USA) (32/46). SpO_2_ data were collected with two sampling frequencies of 1 and 0.5 Hz and stored for later analysis, and all patient information was replaced by individual research codes. 

Clinical ultrasound recordings were performed using a Philips HD-11XE ultrasound system and probe C8-5. Ultrasounds were reviewed and graded by experienced paediatric radiologists, who were blinded to the patients’ clinical characteristics and haemodynamic findings. Volpe’s criteria [27] were used to classify IVH, and the highest grade of IVH identified by ultrasound in the first week of life was used in a dichotomous outcome label. For the purpose of our machine learning model, brain injury was defined as IVH grade II-IV in the first week of life, and IVH grade I or no IVH was classified as non/mild brain injury for this study [28]. It is important to note that infants that were labelled as IVH grade I in our study did not progress to IVH II, and their haemorrhage often improved within the first week. 

A combination of clinical data, used by clinicians for the early identification of infants at risk of IVH, were analysed in this study. These clinical attributes included the following binary variables: sex, hypotension, receiving any inotrope, and the presence of chorioamnionitis in utero. Other continuous variables included birth weight (g) and head circumference (cm), and discrete variables included Apgar score in the first 5 min of life and gestational age (GA, days).

### 2.3. Signal Processing

The INVOS device records using a nonuniform sampling frequency, typically between 1/5 and 1/6 Hz. At this sampling frequency, these devices need external memory to store the data. When this device was not available, the INVOS device stored the data in internal memory with a nonuniform sampling frequency typically between 1/34 and 1/35 Hz. To generate a uniform sampling rate, cubic spline interpolation was used to fill in the missing data [29]. Then, the rcSO_2_ signal was up-sampled to a sampling frequency of 10 Hz, and a low-pass filter (a zero-phase finite impulse response filter, FIR) was used to prevent aliasing before down-sampling to 1/6 Hz. Missing data that were filled in using cubic spline interpolation were removed. 

In some cases, the NIRS probe became loose or detached or was temporarily removed during recordings over 72 h. In these cases, rcSO_2_ was not recorded or saturated at the minimum value of 15% [30]. Thus, in data points equivalent to 15%, a collar of 30 s was applied, and those data points were removed.

To preprocess the SpO_2_ signals, first values of less than 20% or sudden changes (more than 4% in one second) with a collar of 30 s were removed [31]. Missing data were interpolated using cubic spline interpolation [29], and the signal was down-sampled, after applying an appropriate anti-aliasing low-pass filter, to 1/6 Hz for agreement with the rcSO_2_ signal. Missing data prior to down-sampling was then removed. 

### 2.4. Extracting Prolonged Relative Desaturations

A decomposition method, designed specifically to extract PRDs from rcSO_2_ signals in preterm infants [32], was used to extract PRDs from both the rcSO_2_ and SpO_2_ signals. The method applies a discrete cosine transform to the signals before using singular spectrum analysis to extract the transient-like components [24]. This PRD decomposition method is a data-driven operation that we have shown to be effective in isolating the PRDs from synthesised NIRS-like signals [32]. The duration of these PRDs typically lies in the 2-to-15-min range, and these desaturations have been defined and excluded using data-driven methods rather than using an absolute value as a threshold. 

Three different modalities of the rcSO_2_ were analysed: the rcSO_2_ unprocessed, the rcSO_2_ signal without the PRD component, and the rcSO_2_ PRD component alone. For the first two modalities (i.e., rcSO_2_ and rcSO_2_ without the PRDs), the following methods were employed. First, the signals were filtered using a filter bank according to a dyadic frequency response. Five zero-phase finite-impulse response filters—with a frequency bandwidth of a/(2^b^), where a = f_s_ /2 and b = 0,1,2,3,4—were used for band-pass filtering. Next, quantitative features were extracted for each filtered signal over a 4 h epoch with a 50% overlap. The number of epochs varied between 14 and 37, depending on the duration of recorded NIRS. 

### 2.5. Feature Extraction

Amplitude-modulation features included the mean, standard deviation (SD), the 5th and 95th percentiles of the envelope, skewness, and kurtosis of the signal. Instantaneous frequency (IF) was estimated using a central-finite difference of the phase of the signal; mean, SD, skewness, kurtosis, and the 5th and 95th percentiles of the IF were again used as features. Fractal dimension (FD) was estimated using the Higuchi method. The postnatal age of the infants in each epoch was added to the feature set, resulting in a total feature set of 66 features. 

A distinct set of features were measured directly from the extracted PRDs: frequency of the PRDs per hour, total power, envelope summarised in mean and SD, Hjorth parameters (activity, mobility, and complexity), interspike interval summarised in mean and standard deviation, time spent below 63% rcSO_2_ (85% for SpO_2_ transients), the average of the nadir amplitude of PRDs, the average of the downward slope (the slope of the line between the PRD’s baseline and nadir), the average of the upward slope (the slope of the line between the PRD’s nadir and baseline), and averaged duration of the PRDs. The total feature set for PRDs contained 14 features.

The entire process, from signal decomposition to feature extraction, was also applied to the SpO_2_ signal. 

### 2.6. Machine Learning Models

A predictive model of brain injury was developed using a sequential ensemble of decision trees, known as an extreme gradient boosting machine (XGBoost). To reduce variance in the model, a common problem with small noisy data sets, regularisation was increased from default values by setting the maximum tree depth to 3. The total number of trees was 50, and the learning rate was 0.1. The feature set of 66 features per each 4 h epoch was used to train and test the model, using a leave-one-baby-out cross-validation procedure. In this method, the features derived from one infant were left out, and the remaining data were used to make the model, and then the model was tested on the excluded infant. This process was repeated for all the infants in the study, a total of 46 times, to overcome overfitting. 

This approach was used to develop separate models for the rcSO_2_ and SpO_2_ signals, and then again for extracted PRD components. Feature importance was estimated for significant models, to examine how each feature contributed to the model’s prediction.

Due to its superior performance on nosier data sets, a bagging ensemble known as a random forest was used to develop a predictive model using the clinical attributes. The random forest combines multiple decision trees developed independently on different subsets of the data. The default parameters of the random forest were used with 500 trees. Similarly, a leave-one-baby-out cross-validation procedure was used to train and test the model. 

### 2.7. Combining Models

To test if clinical information could enhance the predictability of detecting brain injury, the rcSO_2_ model and clinical model were combined using a late-stage fusion approach. Specifically, the probabilities generated from the rcSO_2_ model were combined with the probability of the clinical-feature model using the geometric mean. The same approach was applied to combining the SpO_2_ and clinical models. All the steps taken to process the signal, extract the PRDs and features, and utilise the machine learning model are presented in Figure 1. 

### 2.8. Statistical Analysis

Fisher’s exact test was used to compare the proportion of infants with brain injury between the two rcSO_2_ sampling frequencies (~1/6 and ~1/35 Hz). To evaluate the ability of each model to detect brain injury, the area under the receiver operating characteristic curve (AUC) and its corresponding 95% confidence interval (CI) were analysed. AUC was considered significant if the value of AUC and the lower CI were greater than 0.5. An AUC value of 0.5 indicates that the model performs no better than a chance in distinguishing between true and false positives, so any value greater than 0.5 is an improvement over random guessing [33]. CIs were calculated using a bootstrapping procedure with resampling (with replacement) on a per-infant basis. In SpO_2_ XGBoost models, a hyperparameter, scale_pos_weight, was added and set to the ratio of the number of negative to positive samples to account for class imbalance in the dataset. The threshold used on the model probability for calculating sensitivity, specificity, and accuracy was 0.4; thus, any predicted probability scores greater than or equal to 0.4 were considered positive predictions. Matthew’s correlation coefficient (MCC) was also used as a performance metric of the classification models. Additional analysis was performed to calculate the AUC of the separate variables of PRDs, to test how each variable performs in distinguishing between outcome labels. Furthermore, the use of time spent below threshold values (<63% rcSO_2_, <85% SpO_2_) in the predictability of brain injury was examined. This analysis was conducted to compare the efficacy of our machine learning models, which utilises numerous quantitative features, with that of current threshold-based methodologies. The grand average of all extracted PRDs from IVH and non/mild IVH groups was calculated and reported in Figure 2 to illustrate the conformation of the PRDs in each group.

## 3. Results

There were 46 infants born less than 28 weeks of GA with rcSO_2_ recorded for >24 h. The clinical characteristics of infants included in this study are detailed in Table 1. There was no significant association between sampling frequency and outcome indicating that sampling frequency was not a confounding factor (*p* = 0.505, Fisher’s exact test). The average duration of the rcSO_2_ recordings and deleted data due to artefact removal were 60.98 h and 0.004 h, respectively. Averaged duration of recorded SpO_2_ and deleted data due to artefact removal were 63.79 h and 1.49 h, respectively. 

Neither a single clinical attribute nor the combination of all attributes in our model was predictive of the short-term clinical outcome: the random forest model had a median (95% CI) testing AUC of 0.57 (0.39–0.76) (Table 2). Thus, we did not find evidence that clinical data, individually or combined, can detect IVH in the first week after birth. 

The rcSO_2_ models before and after PRD removal were not significantly associated with IVH injury (AUC 0.53, CI = 0.31–0.74 and AUC = 0.54, CI = 0.33–0.74, respectively). Combining these rcSO_2_ models with the clinical model did not improve the performance of the models (Table 2). However, the model using the isolated rcSO_2_ PRDs was significantly associated with brain injury, with an AUC of 0.85 (0.72–0.95). The sensitivity and specificity of this model were 0.85 and 0.58, respectively. Matthew’s correlation coefficient (MCC) was 0.57. Adding the clinical model to the rcSO_2_ PRD model showed no practical improvement in performance (AUC = 0.86 (0.74–0.96), sensitivity = 0.94, specificity = 0.25, MCC = 0.61) (Table 2). For comparison with the current literature, we also tested if the threshold value of 63%, i.e., the time rcSO_2_ spent below 63%, was a useful predictor, but this marker was not predictive of brain injury in our cohort (AUC 0.59, CI = 0.40–0.78) (Table 2).

Estimating feature importance in the rcSO_2_ PRD model showed that a range of features contributed to the XGBoost model, with the top three features being the mean amplitude and negative and positive slope of the PRDs (Figure 3). 

Following analysis of the individual features extracted from rcSO_2_ PRDs, we report that only mean amplitude was significantly associated with IVH with an AUC of 0.73 (95% CI: 0.53–0.92) (Table 3), which was less robust than the XGBoost rcSO_2_ PRDs model. The grand average of the extracted PRDs among all infants in the IVH group vs. non/mild IVH group is presented in Figure 2. A higher amplitude of deflection from the baseline in the IVH group compared with the non/mild IVH group is evident. SpO_2_ was recorded for 32 infants. None of the three SpO_2_ models were significantly associated with brain injury (Table 2). Combination with the clinical model did not improve the detection of brain injury. For comparison with the current literature, we also tested if the threshold value of 85%, i.e., the time SpO_2_ spent below 85%, was a useful predictor, but this marker was not predictive of brain injury in our cohort (AUC 0.52, CI = 0.28–0.76) (Table 2).

## 4. Discussion

This study set out to examine the potential utility of machine learning models of rcSO_2_ signals recorded in the first 72 h after birth in extremely preterm infants and their usefulness in detecting IVH in the first week after birth. The results of this study demonstrate that a data-driven combination of extracted features from isolated PRDs of rcSO_2_ signal was associated with IVH in this cohort of extremely preterm infants. 

Previous studies have employed different methods in interpreting NIRS data as an early predictor of IVH. The absolute value of the NIRS signal is used as a criterion for intervention in some NICUs; however, the NIRS values are variable with gestational age and are dependent upon different algorithms incorporated into the devices [21]. The absolute value of NIRS, predictive of adverse outcome, ranged from <50% [20,21] to <65% [22] with varying sensors. SafeBoosC-III phase 3 of the randomised clinical trial used cerebral oximetry during the first 72 h after birth to guide intervention in extremely preterm infants. The authors report that the use of cerebral oximetry did not alter the incidence of death or severe brain injury at 36 weeks’ postmenstrual age after birth compared with infants assigned to receive usual care [34]. In this trial, if infants randomised to the NIRS group had cerebral oxygenation below the threshold for hypoxia, treatment was considered [35]. Of note, the time rcSO_2_ spent below 63% was not predictive of brain injury in the current preterm cohort (AUC 0.593 CI 0.399–0.775). The potential uncertainty relating to absolute NIRS values necessitates the development of alternative methods, including the use of other features of the signal for their potential to predict outcomes.

The results of the current study indicate that features we extracted from the rcSO_2_ signal, before and after excluding PRDs, were not predictive of IVH. These findings are in contrast with previous signal processing methods that proposed an improvement in performance when the PRDs were removed [24]. However, this study covered a wider GA (<32 weeks), used shorter epochs (2 h), had a different sampling frequency (1/35 Hz) compared with what we used (1/6 Hz), and was limited by a small sample size (*n* = 10). Interestingly, however, in this current study, features arising from the rcSO_2_ PRDs were associated with IVH. If validated, AI recognition of this novel feature of NIRS signal during bedside monitoring could facilitate the timely implementation of neuroprotective strategies for the infant.

The PRDs were extracted from the NIRS signal using a previously described method; this procedure is a data-driven tool that identifies transient-like waveforms of the order of one to tens of minutes within a signal. Therefore, the PRDs are not predefined events determined by absolute amplitude, duration, or frequency. Post-extraction analysis of the PRDs, as illustrated in Figure 2, indicates that they are events lasting minutes with a relative change in signal of approximately 11–14%. Features of these estimated PRDs are then combined using a machine learning model, a process that can detect variances not discernible visually and also a process that due to its complexity can be difficult to relate back to signal characteristics.

This study contributes to the recent debate about the role of rcSO_2_ in predicting early outcomes. We must strive to overcome current limitations in the interpretation of NIRS with the use of innovative analysis. The quantification of rcSO_2_ is considered to arise from venous/arterial blood (70%/30%) and is influenced by many potential variables, including perfusion, cerebral autoregulation, cerebral activity, and oxygen extraction, which makes it more challenging to interpret [36]. Previous studies have reported a strong correlation between cerebral oxygen desaturation and low cardiac output in preterm infants [37,38]. The continuous and simultaneous measurement of mean arterial blood pressure, cardiac output, and EEG brain activity could provide additional insight if incorporated into the analysis of rcSO_2_.

SpO_2_ has been used to guide respiratory care. In 2007, the American Academy of Pediatrics recommended a range of 85–95% in preterm infants, and in 2010 the recommended target range was set to 85–93% by the European Association of Perinatal Medicine [39,40]. More recently, large clinical studies, such as NeOProM (Neonatal Oxygenation Prospective Meta-analysis) have narrowed the range of SpO_2_ [41]. Pooled data from that clinical trial reported that extremely preterm infants with SpO_2_ values of 85–89% were at higher risk of mortality and necrotizing enterocolitis compared with ones with SpO_2_ values of 91–95%. However, the risk of retinopathy of prematurity was higher in preterm infants with higher levels of SpO_2_ [41]. Sullivan et al. [42] suggested that the combined features of SpO_2_ (mean, SD, kurtosis, and skewness) with clinical data in the first 12 h and the first 7 days of life were associated with severe IVH (grade III and IV) and NEC. In terms of absolute values, Vesoulis et al. [43] studied 645 infants born <32 weeks of GA and proposed that the time spent with SpO_2_ ≤ 70% was associated with severe grade III and IV IVH. Moreover, a lower level of SpO_2_ was associated with IVH grades III and IV in very low birth weight infants [44]. However, in the current study, we did not find an association between features extracted from the SpO_2_ signal, including and excluding PRDs or with time spent below SpO_2_ ≤ 85%, with IVH outcome. If the isolated PRDs in the rcSO_2_ were caused by systemic hypoxemia, we reasoned that PRDs isolated from the SpO_2_ signal would also be predictive of IVH outcome. This would be significantly advantageous as the use of pulse oximetry is more widespread than cerebral oxygen monitoring. However, we did not find features of isolated PRDs from the SpO_2_ signal that were associated with IVH injury.

SpO_2_ is dependent on detecting a variance in transmitted light. Thus, when perfusion decreases, pulse amplitude gets smaller and pulse oximetry readings are adversely affected. As a result, hypotension and hypothermia that cause poor peripheral perfusion can render pulse oximetry prone to error [45]. Furthermore, pulse oximetry is influenced by the presence of foetal haemoglobin (FHb) [46]. FHb is the main oxygen carrier during pregnancy and starts to be replaced by adult haemoglobin from the 20th week of gestation [47]. FHb has a higher affinity for oxygen compared with adult haemoglobin and does not provide enough oxygen diffusion and delivery in neonates. Thus, the probability of higher concentration of FHb and increased affinity for oxygen and the fact that 20 out of 46 infants were hypotensive might contribute to the poor association between SpO_2_ and IVH outcome in our cohort.

PRDs were evident in all infants and were extracted from both rcSO_2_ and SpO_2_. Further investigation in a subset of infants revealed that there were both asynchronous and synchronous PRDs in both signals. These events in SpO_2_ may represent systemic hypoxemia brought about by poor gas exchange or decreased perfusion. Independent decreases in rcSO_2_ may reflect selective overt impairment of regional cerebral perfusion or instances of increased oxygen utilisation in the face of enhanced functional demand with inadequate neurovascular coupling.

A limitation of this study is that the outcome was defined based on the highest grade of IVH diagnosed using ultrasound in the first week of life, and the exact time of the IVH occurrence was unknown. Furthermore, SpO_2_ was recorded for only 32 out of 46 infants, and SpO_2_ and rcSO_2_ were time-synchronised for only 26 infants. Moreover, the signal decomposition method used to extract the PRDs performed well in isolating the desaturations that were long (average of 8 min), but many shorter (approximately <1 min) desaturation events were not captured using this method. Algorithms could be developed to isolate these short transients and test their predictability of IVH in future studies. 

This study proposes that the features extracted from the rcSO_2_ signal contain information that may be useful in the early detection of brain injury, which may not be the case for the SpO_2_ signal. Decreases in cerebral oxygen saturation may reflect hypoxia, vasoconstriction, low blood flow, and/or impaired autoregulation. It is not possible to determine the individual or combined elements of each of these possibilities. However, what is clear is that a combination of quantitative features of these relative cerebral desaturations (PRDs) are associated with brain injury in extremely preterm infants. The incorporation of machine learning methods needs to be validated in future large cohorts of extremely preterm infants. 

## Figures and Tables

**Figure 1 children-10-00917-f001:**
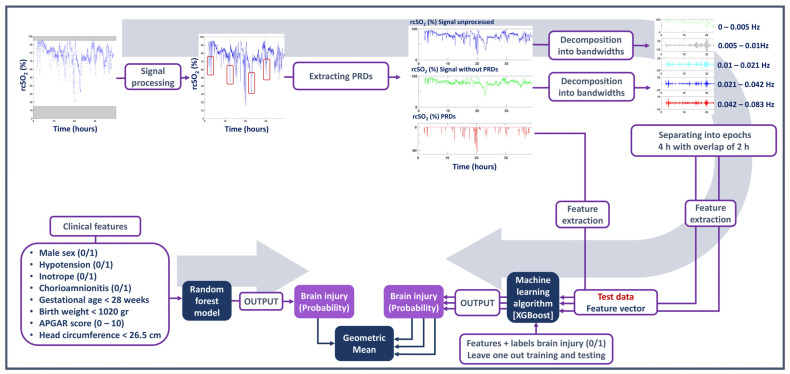
Summary of the steps taken to process the signal, extract the prolonged relative desaturations (PRDs) and features, and utilise the machine learning model.

**Figure 2 children-10-00917-f002:**
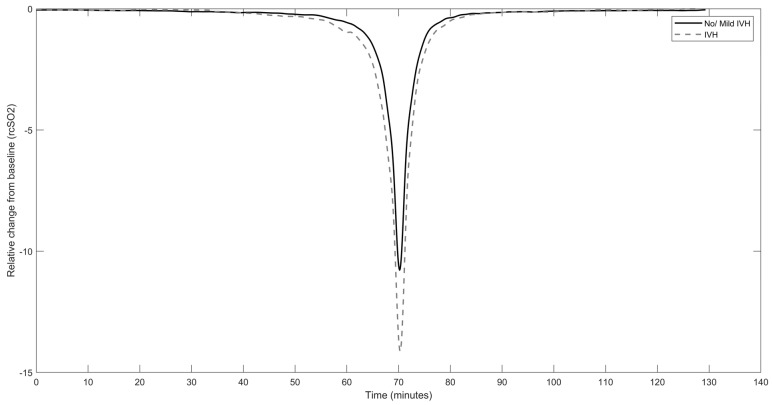
Averaged curve of all rcSO_2_ PRDs in the intraventricular haemorrhage group (**dashed line**) vs. no/mild intraventricular haemorrhage group (**solid line**); X-axis indicates the time in minutes when the grand average of rcSO_2_ PRDs starts to deviate from baseline, reaches the nadir, and increases to baseline again.

**Figure 3 children-10-00917-f003:**
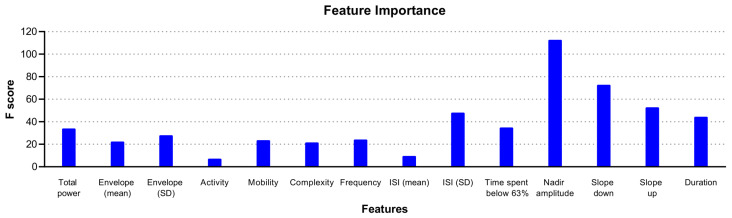
Illustration of feature importance of the XGBoost model using rcSO_2_ prolonged relative desaturations (PRDs) in predicting intraventricular haemorrhage.

**Table 1 children-10-00917-t001:** Clinical data of the subset of infants from the HIP trial included in the current study.

Clinical Features	Brain Injury (*n* = 12)	No/Mild Injury (*n* = 34)
Sex (male) (% of the total number of infants in the group)	8 (67%)	17 (50%)
Hypotension (% of the total number of infants in the group)	7 (58%)	13 (38%)
Inotrope administration (% of the total number of infants in the group)	3 (25%)	6 (18%)
Chorioamnionitis (% of the total number of infants in the group)	0 (0%)	5 (15%)
Gestational age (days) Mean (SD)	178.5 (7.5)	181.3 (8.8)
Birth weight (g) Mean (SD)	769.2 (173.7)	766.8 (151.5)
APGAR score (5 min) Mean (SD)	7.0 (2.0)	6.9 (1.9)
Head circumference (cm) Mean (SD)	23.3 (1.5)	23.3 (1.5)

**Table 2 children-10-00917-t002:** Area under the curve (AUC) values for the signals used to predict brain injury in the first week of life.

Model	AUC	95% Confidence Interval
Clinical data	0.575	0.390–0.759
rcSO_2_	0.532	0.312–0.740
rcSO_2_ without PRDs	0.542	0.331–0.741
rcSO_2_ PRDs	0.846 *	0.720–0.948
rcSO_2_ and clinical data	0.571	0.390–0.756
rcSO_2_ without PRDs and clinical data	0.591	0.396–0.777
rcSO_2_ PRDs and clinical data	0.860 *	0.742–0.957
SpO_2_	0.198	0.000–0.459
SpO_2_ without PRDs	0.261	0.061–0.545
SpO_2_ PRDs	0.493	0.264–0.741
SpO_2_ and clinical data	0.324	0.133–0.546
SpO_2_ without PRDs and clinical data	0.348	0.151–0.579
SpO_2_ PRDs and clinical data	0.454	0.235–0.697
rcSO_2_ time spent below 63%	0.593	0.399–0.775
SpO_2_ time spent below 85%	0.522	0.277–0.759

* indicates significant AUC.

**Table 3 children-10-00917-t003:** Area under the curve (AUC) values for the individual features of rcSO_2_ PRDs used to predict brain injury in the first week of life.

Feature	AUC	95% Confidence Interval
Frequency of the transients per hour	0.409	0.201–0.619
Total power	0.368	0.191–0.560
Envelope (mean)	0.397	0.214–0.592
Envelope (standard deviation)	0.539	0.352–0.738
Hjorth parameters (activity)	0.603	0.399–0.802
Hjorth parameters (mobility)	0.409	0.214–0.602
Hjorth parameters (complexity)	0.569	0.353–0.775
Inter-spike interval (mean)	0.532	0.306–0.748
Inter-spike interval (standard deviation)	0.505	0.282–0.714
Time spent below 63%	0.691	0.494–0.861
Nadir amplitude of PRDs (mean)	0.733 *	0.530–0.919
Slope down (mean)	0.689	0.474–0.883
Slope up (mean)	0.669	0.477–0.838
Duration of the PRDs (mean)	0.603	0.422–0.770

* indicates significant AUC.

## Data Availability

The data used and analyzed during the current study involve sensitive patient information.

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
