# Peer review of "Machine Learning Detects Intraventricular Haemorrhage in Extremely Preterm Infants [Author-notes fn1-children-10-00917]"

_children, 2023, doi:10.3390/children10060917_

Round 1
Reviewer 1 Report
Several aspects of yot study need immediate improvement.
First - the description of the studied population lacks at the beginning of the paper....we find about the population characteristics only in the results section.
Second- the birth complications of the patients included in the study should be studied more thoroughly....because they might contribute to the brain lessons
Third- there are some errors regarding the references inside the paper
Fourth - there should be a more detailed description of the ultrasound machine and the type of probe used for ultrasound evaluation
English is good.....needs only minor corrections
Reviewer 2 Report
Machine Learning Detects Intraventricular Haemorrhage in Extremely Preterm Infants
In this study the authors aim to test the potential utility of applying machine learning methods to regional cerebral (rcSO2) and peripheral oxygen saturation (SpO2) signals to detect brain injury in extremely preterm infants. They also wanted to examine the potential value of combining clinical features with features from the NIRS signal to detect preterm brain injury. They study a subset of infants enrolled in the Management of Hypotension in Preterm infants trial, <28 weeks gestational age, with continuous rcSO2 measurements performed over the first 72 hours and cranial ultrasounds performed during the first week after birth. They used a machine learning model to detect brain injury (IVH grade II-IV) and found that there was a significant association between data-driven definition of prolonged relative desaturations (PRDs) in rcSO2 and brain injury. They conclude that automated analysis of PRDs of the cerebral NIRS signal in extremely preterm infants may aid, better prediction of IVH compared to a threshold-based approach.
The study is well written and the results are very interesting.
Just one question out of personal curiosity related to the results shown in Table 2. As far as I understand, the range of the AUC should be between 0.5 (similar to random) and 1 (100% sensitivity and specificity). Could you please explain the meaning of an AUC less than 0.5? Can a test be "worse" than chance? In such a case, wouldn't it be "good to discard"? I am a bit confused about this.
A “Table 3” is referred several times in the text (lines 255, 259 and 280), but there is no such Table in the manuscript. Is Table 2 meant? Please, correct it.
Minor.
Please, check reference 17 (line 110).
Line 148. Reference (1720). Please, correct it.
Line 311. References (2325, 26). Please, correct it.
Round 2
Reviewer 1 Report
I noticed that more statistical data were introduced in the original text but I also noticed that there is a lack of sufficient explanation about the statistical data newly introduced.
So I suggest more detailed info about the statistics should be introduced.
The overall work isto be appreciated but there is still a need for improvement.
Please be more specific about the statistics and verify again the grammar of the text.
Just minor improvements s needed
Author Response
We would like to thank you for your considered feedback which has improved the coherence of the manuscript.
We have revised the ‘Statistical Analysis’ section of the methods which now appears as follows;
Statistical Analysis
Fisher’s exact test was used to compare the proportion of infants with brain injury between the two rcSO2 sampling frequencies (~1/6 and ~1/35 Hz). To evaluate the ability of each model to detect brain injury, area under the receiver operating characteristic curve (AUC) and its corresponding 95% confidence interval (CI). AUC was considered significant if the value of AUC and the lower CI were greater than 0.5. An AUC value of 0.5 indicates that the model performs no better than chance in distinguishing between true and false positives, so any value greater than 0.5 is an improvement over random guessing [33]. CIs were calculated using a bootstrapping procedure using resampling (with replacement) on a per-infant basis. In SpO2 XGBoost models, a hyperparameter, scale_pos_weight was added and set to the ratio of the number of negative to positive samples to account for class imbalance in the dataset. The threshold used on the model probability for calculating sensitivity, specificity and accuracy was 0.4, thus any predicted probability scores greater than or equal to 0.4 were considered a positive prediction. Matthew’s correlation coefficient (MCC) was also used as a performance metric of the classification models. Additional analysis was performed to calculate the AUC of the separate variables of PRD’s, to test how each variable performs in distinguishing between outcome labels. Furthermore, the use of time spent below threshold values (< 63% rcSO2, <85% SpO2) in predictability of brain injury was examined. This analysis was conducted to compare the efficacy of our machine learning models, which utilizes numerous quantitative features, with that of current threshold-based methodologies. The grand average of all extracted PRDs from IVH and non/mild IVH groups were calculated and reported in Figure 3 to illustrate the conformation of the PRDs in each group.
We have also sought to improve the clarity of the text by making other minor changes in the methods (line 214) and results section (line 306).
In order to enrich the literature of our manuscript we have cited more scholars in the field in the introduction of the manuscript (line 58-73) and in the discussion (line 394) and in doing so have decreased the rate of self-citation.
The papers we have cited by our own group include methodological papers which we believe are necessary to support the comprehension of the work (Dempsey et al., 2021, O’Toole et al., 2016, 2018, Ashoori et al., 2021) and others which provide a strong rationale for the current study (Alderliesten et al., 2016, van Bel et al 2008). Other citations which have been kept to a minimum are significant studies in the field, which authors of this manuscript are contributory, but not lead authors (SafeBoosC Trial Plomgaard et al., 2019, Hansen et al 2023).